



# Technical note: On the reliability of laboratory beta-source calibration for luminescence dating

**Barbara Mauz**[1], **Loïc Martin**[2], **Michael Discher**[1], **Chantal Tribolo**[2], **Sebastian Kreutzer**[3,2], **Chiara Bahl**[1], **Andreas Lang**[1], **and Nobert Mercier**[2]

[1]Department of Geography and Geology, University of Salzburg, 5020 Salzburg, Austria
[2]IRAMAT-CRP2A, UMR 5060, CNRS-Université Bordeaux Montaigne, 33600 Pessac, France
[3]Geography & Earth Sciences, Aberystwyth University, Aberystwyth SY23 3FL, UK

**Correspondence:** Barbara Mauz (barbara.mauz@sbg.ac.at)

**Abstract.** The dose rate of the $^{90}$Sr $/\,^{90}$Y beta source used in most luminescence readers is a laboratory key parameter. There is a well-established body of knowledge about parameters controlling accuracy and precision of the calibration value but some hard-to-explain inconsistencies still exist. Here, we have investigated the impact of grain size, aliquot size and irradiation geometry on the resulting calibration value through experiments and simulations. The resulting data indicate that the dose rate of an individual beta source results from the interplay of a number of parameters, most of which are well established by previous studies. Our study provides evidence for the impact of aliquot size on the absorbed dose in particular for grain sizes of 50–200 μm. For this grain-size fraction, the absorbed dose is enhanced by $\sim 10\,\%$–20 % as aliquot size decreases due to the radial increase of dose rate towards the centre of the aliquot. The enhancement is most variable for 50–100 μm grains mounted as aliquots of < 8 mm size. The enhancement is reversed when large grains are mounted as small aliquots due to the edge effect by which the dose induced by backscattered electrons is reduced. While the build-up of charge dictates the increase of absorbed dose with the increase of grain size, this principle becomes more variable with changing irradiation geometry. We conclude that future calibration samples should consist of subsamples composed of small, medium, large and very large quartz grains, each obtaining several gamma doses. The calibration value measured with small, medium and large aliquots is then obtained from the inverse slope of the fitted line, not from a single data point. In this way, all possible irradiation geometries of an individual beta source are covered, and the precision of the calibration is improved.

## 1 Introduction

The dose rate of the $^{90}$Sr $/\,^{90}$Y beta source used in most luminescence readers is a laboratory key parameter. If the source's calibration is incorrect, results for equivalent dose and age are also incorrect. The significance of beta-source calibration is therefore well known and has been subject to interlaboratory comparison studies (e.g. Pernicka and Wagner, 1979; Göksu et al., 1995).

Past studies have established that charge build-up, attenuation and backscatter constitute the physical mechanisms controlling the dose absorbed in the sample's mineral grain. The interplay of these mechanisms depends on mineral type (Aitken, 1985), on grain transparency (Bell and Mejdahl, 1981), beta-source-to-grain distance (Wintle and Aitken, 1977), grain size (Goedicke, 2007; Armitage and Bailey, 2005; Mauz and Lang, 2004) and the sample carrier's substrate (Greilich et al., 2008; Armitage and Bailey, 2005; Mauz and Lang, 2004; Wintle and Aitken, 1977). In addition, accuracy and/or precision of the calibration value depend on the measurement protocol (Guérin and Valladas, 2014; Kadereit and Kreutzer, 2013), the atomic numbers ($Z$) of mineral and sample carrier (Hansen et al., 2018) and the accuracy of the gamma dose to mineral calculation (Burbidge et al., 2016; Tribolo et al., 2019). Despite this well-established body of knowledge, Hansen et al. (2015) note

an unexplained 3 % dispersion of their calibration data, subsequently investigated by Autzen et al. (2017). They show that overdispersed calibration data result from attenuation and backscattering, which change in response to changing grain shape and changing sample-carrier material (Autzen et al., 2017). As a consequence, the beta-dose rate should decrease for grain sizes > 100 µm (Wintle and Aitken, 1977) because with increasing grain size the contribution of low-energy backscatter decreases and the primary energy spectrum is more attenuated (Hansen et al., 2018; Greilich et al., 2008). While this has improved our understanding of calibration data significantly, some details are still not fully explained. Here, we test the hypothesis that, in addition to grain size and disc substrate, aliquot size and beta-source shape influence the dose rate. We carried out experiments using three quartz calibration samples characterised by three different grain-size fractions arranged in aliquots of different sizes and compared the experimental data with simulated data obtained from *GEANT4* (Agostinelli et al., 2003) and *MCNP6* (Werner, 2017; Werner et al., 2018). The results from experiments should allow identifying the impact of grain size, aliquot size and beta-source shape on the dose rate. The simulations should provide a more complete picture of the impact of individual parameters that is hard to achieve with experimental data due to experimental uncertainties being typically above 5 %. We show here that grain size and aliquot size impact on the absorbed dose in response to the irradiation geometry and that this interplay should be reflected in the design of calibration measurements.

## 2 Experimental details

### 2.1 Luminescence readers and beta sources

For all experiments $^{90}Sr/^{90}Y$ beta sources with $E_{max} =$ 2.26 MeV (Aitken, 1985) built in, three different *lexsyg* luminescence readers of Freiberg Instruments were used. One is the *lexsyg RESEARCH* reader (Richter et al., 2013) equipped with a beta source arranged in a ring of 17 sealed "mini-sources" with a nominal activity of 1.51 GBq. The other two readers are *lexsyg SMART* readers (Richter et al., 2015): one is equipped with a planar beta source and the other is equipped with a ring composed of 23 mini-sources, both with a nominal activity of 1.85 GBq. The *SMART* ring-shaped source is closed at the top (hereafter named "closed ring"), while it is open in the *RESEARCH* (hereafter named "open ring") to allow for radio-fluorescence measurements.

The ring-shaped sources consist of mini-sources. For the open-ring source, these mini-sources were tested for homogeneous activity (< 5 % variation; Richter et al., 2012). As a result, the radiation field of this source varies by 2 %–8 % across an 8–10 mm cup diameter (Richter et al., 2012). The larger variation occurs towards the cup edge due to increasing backscatter from the cup rim, but the inner 6 mm of the cup are exposed to a very homogeneous radiation field

(Richter et al., 2012). The sources of the *SMART* readers are not pre-selected for homogeneous activity and may deliver a less uniform radiation field. With a distance of 6.9 mm between source and sample-holder surface, the radiation field of all sources is expected to be curved. Veronese et al. (2007) show that the dose-rate reduction follows a power function which yields a parabolic curve of variable width. A very wide, and hence flat, parabolic curve is delivered by the open-ring source (Richter et al., 2012) due to its special design.

### 2.2 Calibration samples

Samples used for the experiments are listed in Table 1. In terms of grain size, the samples fall in two categories: (1) fine-grain aliquots are composed of 4–11 µm grains and are always 7.95 mm in size; (2) coarse-grain aliquots are composed of 180–250 or 90–160 µm grains and can be of small (1 mm), medium (3 mm) and large (5–7.95 mm) aliquot size. The Risø fine-grain sample (batch no. 108) is described in Hansen et al. (2015). The Freiberg coarse-grain sample is described in Richter et al. (2020). Tribolo et al. (2019) report on gamma irradiation and calculation of absorbed gamma dose.

### 2.3 Sample carrier

To limit the complexity of the study, only one type of sample carrier was used in our experiments. The sample carrier is a cup (Fig. 1) with dimensions varying by up to 0.1 mm (our own measurements of 10 cups). The cup is made of standard stainless steel ("stainless steel 1.4841"; short name: X15CrNiSi25-21) with a chemical composition of C ($\leq 0.20$ %), Si ($\leq 1.5$ %–2.5 %), Mn ($\leq 2.00$ %), P ($\leq 0.045$ %), S ($\leq 0.015$ %), Cr (24.00 %–26.00 %), Ni (19.00 %–22.00 %), N ($\leq 0.11$ %) and Fe (> 50 %). The material is heat resistant up to approximately 1150 °C (e.g. https://www.thyssenkrupp-materials.co.uk/stainless-steel-314-14841.html, last access: 7 June 2021).

### 2.4 Measurement protocol

A standard single-aliquot regenerative (SAR) dose protocol was employed with irradiation doses adjusted to encompass the expected interpolation point on the dose-response curve and test doses typically around 10 % of the expected interpolation point (in seconds). The stimulation power of the blue LEDs (458Δ5 nm) was reduced as aliquot size increased to avoid overexposure of the photomultiplier. The efficiency of the protocol was tested using undosed subsamples (dose recovery better than 5 %; Tribolo et al., 2019). The measurement parameters are listed in Table 2.

**Table 1.** Samples and their codes used in the experiments. DTU Nutech: Center for Nuclear Technologies, Denmark; SSDL: Secondary Standard Dosimetry Laboratory, Munich. For SSDL calibration samples, the absorbed gamma dose and its uncertainty are derived from a Monte Carlo (MC) simulation. The uncertainty of the dose (2.1 %) is the quadrature of errors resulting from the MC simulation (1.4 %), from the air kerma (1 %) and from the geometry of the irradiation field (1.2 %); see also Table 2 in Tribolo et al. (2019). For DTU calibration samples, the calculation was revised (Martin Autzen, personal communication, December 2019).

| Sample | Sample code | Grain size (µm) | Grain size in practical terms | $\gamma$ dose (Gy) | $\gamma$-dose lab |
| --- | --- | --- | --- | --- | --- |
| Risø batch no. 17 | R17_180 | 180–250 | Coarse grain | $5.0 \pm 0.1$ | DTU |
| Risø batch no. 113 | R113_180 | 180–250 | Coarse grain | $5.0 \pm 0.1$ | DTU |
| Risø batch no. 108 | R108_4 | 4–11 | Fine grain | $5.0 \pm 0.1$ | DTU |
| Freiberg-2019 | F19_90 | 90–160 | Coarse grain | $3.00 \pm 0.06$ | SSDL |
| Freiberg-2014 | F14_90 | 90–160 | Coarse grain | $3.00 \pm 0.06$ | SSDL |

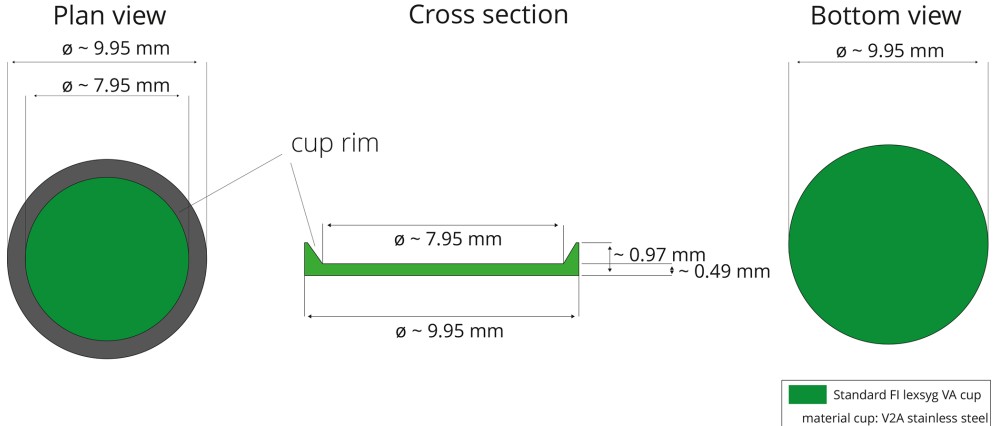

**Figure 1.** The shape of the stainless-steel sample carrier (cup) used in the *lexsyg* readers.

## 3   Simulation details

The simulation of the irradiation in the *lexsyg SMART* was performed using the *GEANT4* and *MCNP6.2* toolboxes. The irradiation geometry simulated (Fig. 2) was adopted from the technical description of the manufacturer and from the sample-carrier description (Fig. 1) with the sample placed in the centre of the cup. Source and housing including the fixing screws were represented as one stainless-steel cylinder surrounded by a stainless-steel shield. The quartz grains were not considered individually but represented as a cylinder, the size of which was modified according to the grain size (height) and aliquot diameter to be simulated. For simulating the dose rate as a function of depth in a given aliquot, the "sample cylinder" was subdivided into 5 or 10 µm thick layers depending on the grain size to be modelled. The material was $SiO_2$ with a density of $1.8\,g\,cm^{-3}$ which represents the packing of sand- and silt-sized spherical grains mounted as aliquots. A 5 µm layer of silicon oil was added between the sample and the sample carrier for the simulation of coarse-grain aliquots (grain sizes from 25 to 250 µm). The spectra of the $^{90}Sr\,/\,^{90}Y$ beta source were simulated using the *GEANT4* radioactive decay function (Hauf et al., 2013). Then

$10^8$ disintegrations of $^{90}Sr$ were simulated in each run, and three runs were carried out for each aliquot configuration. The Penetration and Energy Loss of Positrons and Electrons (PENELOPE) code for low-energy particle physics (Baró et al., 1995; Ivanchenko et al., 2011) was employed to calculate path and interaction of the beta particle with the structures presented in the model. The dose deposited in the $SiO_2$ target was recorded in the whole sample cylinder, and a dose-rate profile was constructed as a function of depth in the sample. For simulating small aliquots, the *MCNP6* code was used: the target was split into seven spherical cells (Fig. 2b) and the F6 tally was used to simulate the energy deposition averaged over the target cell for electrons and photons separately (see the Supplement for details). The output files produced by the *MCNP6* code were used to quantify photon and electron production originating from the interaction mechanisms between beta particle and matter (for details, see the Supplement). The precision of the *GEANT4*-derived result was calculated for each aliquot configuration at the 95 % confidence level (0.95 CL), based on the standard deviation between the results of the three runs per simulation. The uncertainty of the *MCNP6*-derived result was obtained from the fractional standard deviation calculated by the Monte Carlo routine.

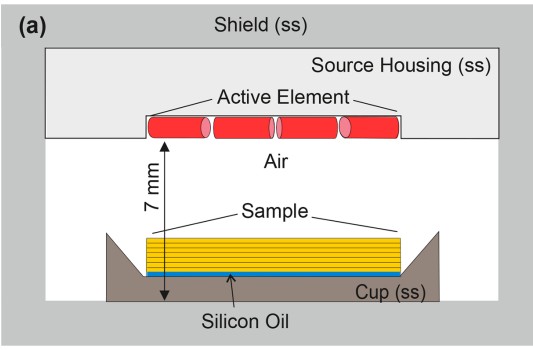 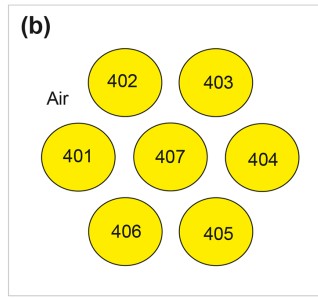

**Figure 2.** The geometry of the $^{90}$Sr / $^{90}$Y source in the *lexsyg SMART* as designed for the simulation. **(a)** The *GEANT4* simulations (not to scale; ss indicates stainless steel). The active element is a ring of 17 small beta sources closed to the top, or it is a planar foil. The cylinder-shaped sample is represented by 5–10 µm thick layers resting on a 5 µm layer of silicon oil (blue colour). The aliquot size illustrated is 7.95 mm. The distance between bottom of cup and surface of source is 7 mm; **(b)** plan view on individual grains represented as spheres of SiO$_2$ used in the *MCNP6* simulations. Cell numbers 401–406 represent "edge grains", and cell number 407 is the central grain.

**Table 2.** Samples, luminescence readers and measurement parameters used in the experiments. To avoid overexposure of the photomultiplier, the stimulation power was 5, 10, 70 and 100 mW cm$^{-2}$ of the blue LEDs (458$\Delta$5 nm) depending on the size of the aliquot; PH/CH: preheat and cut heat temperatures for regeneration and test doses, respectively; preheat was for 10 s. For sample description, see also Table 1.

| Sample | Reader and beta-source geometry | Aliquot size (mm) | $n$ measured | PH/CH (°C) |
|---|---|---|---|---|
| R17_180 | *RESEARCH* open ring | 7.95 | 5 | 260/260 |
| | | 3 | 10 | |
| | | 1 | 4 | |
| F19_90 | *RESEARCH* open ring | 7.95 | 5 | 260/260 |
| | | 3 | 5 | |
| | | 1 | 4 | |
| R108_4 | *RESEARCH* open ring | 7.95 | 10 | 240/200 |
| R17_180 | *SMART* planar | 7.95 | 5 | 260/260 |
| | | 3 | 10 | |
| | | 1 | 4 | |
| F19_90 | *SMART* planar | 7.95 | 5 | 260/260 |
| | | 3 | 5 | |
| | | 1 | 4 | |
| R108_4 | *SMART* planar | 7.95 | 10 | 240/200 |
| R113_180 | *SMART* closed ring | 7.95 | 6 | 230/200 |
| | | 5 | 6 | |
| | | 1 | 6 | |
| F14_90 | *SMART* closed ring | 5 | 6 | 200/200 |
| | | 1 | 4 | |
| R108_4 | *SMART* closed ring | 7.95 | 2 | 240/200 |

## 4 Results

### 4.1 The calibration material

The calibration samples provided by the manufacturers show high sensitivity to the dose and, consequently, excellent reproducibility. Small-to-large differences between samples are evident from the experimental data which are not systematic but seem to depend on measurement parameters (e.g. aliquot size) and, eventually, on the calculation of the gamma dose (Tribolo et al., 2019). In fact, Tribolo et al. (2019) identified an up-to-14 % difference of dose rate between samples when analysing single grains of the same calibration samples used here (F14_90; R113_180; Table 2). This was subsequently reduced as a result of one of the manufacturers changing their gamma-dose calculation which is still subject to ongoing research (Martin Autzen, personal communication, June 2021). TS1

### 4.2 Uncertainty of data

The total uncertainty of the experimental data is derived from the optically stimulated luminescence (OSL) measurement statistics and the uncertainty of the gamma dose amounting to a standard error of the mean of 2 %–4 %. At the 95 % confidence level ($t_{95}$), the uncertainty is around 4 %–7 % for $n > 5$ and 8 %–13 % for $n < 5$ (Table 3) due to the small number of aliquots measured. Therefore, we regard differences between individual dose-rate values of $> 4$ % as informative and differences $> 8$ % as significant. For the *GEANT4*-derived simulation data, the uncertainty is 0.15 %–3.00 %, where the majority of the data show an uncertainty of $< 1$ % due to the expected excellent reproducibility of the simulation runs. The *MCNP* uncertainty is the fractional standard deviation which is typically 0.1 %–1.1 % in our study.

**Table 3.** Beta-dose rates obtained from experiments. Open ring is the beta source of the *lexsyg RESEARCH* reader, planar is the one of the *lexsyg SMART* (built 2017) reader, and closed ring is the one of the other *lexsyg SMART* (built 2014) reader (Fig. 1). Dose rates listed are mean values with uncertainties quoted at 95 % confidence level ($t_{95}$) derived from the $t$ distribution for $n-1$. Mean dose rates were corrected for the decay of the $^{90}$Sr / $^{90}$Y source using $t_{1/2} = 28.79$ years and the time elapsed since reference datum (21 January 2020). Uncertainty of the normalised value is relative to the numerator which is aliquot size; fg indicates fine grain. For details, see the text.

| Beta source | Sample code | Grain size (µm) | $n$ | Aliquot size (mm) | Dose rate (Gy s$^{-1}$) corrected ($t_{95}$) | Dose rate normalised to fg (%) |
|---|---|---|---|---|---|---|
| Open ring | R17_180 | 180–250 | 5 | 7.95 | $0.0617 \pm 0.0028$ | $97.68 \pm 4.47$ |
| | | | 10 | 3 | $0.0592 \pm 0.0023$ | $93.57 \pm 3.62$ |
| | | | 4 | 1 | $0.0633 \pm 0.0030$ | $99.95 \pm 4.70$ |
| Open ring | F19_90 | 90–160 | 5 | 7.95 | $0.0631 \pm 0.0034$ | $99.66 \pm 1.44$ |
| | | | 5 | 3 | $0.0621 \pm 0.0032$ | $98.19 \pm 5.11$ |
| | | | 4 | 1 | $0.0641 \pm 0.0051$ | $101.30 \pm 8.03$ |
| Open ring | R108_4 | 4–11 | 10 | 7.95 | $0.0633 \pm 0.0023$ | $100.00 \pm 3.68$ |
| Planar | R113_180 | 180–250 | 5 | 7.95 | $0.1167 \pm 0.0075$ | $104.62 \pm 6.70$ |
| | | | 10 | 3 | $0.1297 \pm 0.0050$ | $116.24 \pm 4.45$ |
| | | | 4 | 1 | $0.1247 \pm 0.0088$ | $111.76 \pm 7.91$ |
| Planar | F19_90 | 90–160 | 5 | 7.95 | $0.1184 \pm 0.0056$ | $106.31 \pm 5.10$ |
| | | | 5 | 3 | $0.1296 \pm 0.0074$ | $116.19 \pm 6.65$ |
| | | | 4 | 1 | $0.1228 \pm 0.0100$ | $114.93 \pm 8.93$ |
| Planar | R108_4 | 4–11 | 10 | 8 | $0.1116 \pm 0.0043$ | $100.00 \pm 4.10$ |
| Closed ring | R17_180 | 180–250 | 6 | 7.95 | $0.1460 \pm 0.0064$ | $102.10 \pm 4.46$ |
| | | | 6 | 5 | $0.1440 \pm 0.0060$ | $100.70 \pm 4.22$ |
| | | | 6 | 1 | $0.1580 \pm 0.0072$ | $110.49 \pm 5.01$ |
| Closed ring | F14_90 | 90–160 | 6 | 5 | $0.1670 \pm 0.0115$ | $116.78 \pm 8.03$ |
| | | | 4 | 1 | $0.1800 \pm 0.0121$ | $125.87 \pm 8.48$ |
| Closed ring | R108_4 | 4–11 | 2 | 7.95 | $0.1430 \pm 0.0186$ | $100.00 \pm 13.00$ |

## 4.3 Grain size and aliquot size

Our experimental data indicate a grain-size dependence that varies for the coarse-grained samples (90–160 and 180–250 µm) with aliquot size and beta-source geometry between 0 % and 26 % (Fig. S1 in the Supplement and Table 4). The data indicate that the impact of grain size on the dose rates is insignificant for large (7.95 mm) aliquots (Table 4). For aliquot sizes < 7.95 mm, the difference between the two coarse-grained samples is also negligible except for the closed-ring source (Fig. 3). In contrast, the difference between fine-grain and coarse-grain dose rates is 0.4 %–26 % for aliquot sizes < 7.95 mm and the magnitude of this difference is controlled by the individual source (Table 3) and by the distance between source and sample. This latter distance changes with changing grain size resulting in an absorbed dose that is about 3 %–4 % higher for large grains than for fine grains. With decreasing aliquot size, the dose rate increases by ∼ 4 %–8 % for both coarse-grain fractions (Fig. 4), but this increase is statistically not significant (Table 3).

The data obtained from the simulations indicate a rise of dose rate with increasing grain size (Fig. 5). There is a striking similarity between our simulated data and the experimental data adopted from Armitage and Bailey (2005). Indeed, the simulation shows a gradual change of the grain-size effect over the entire grain-size range which is confirmed by the experiment for grain sizes < 55 µm, but for grain sizes > 100 µm the experiment indicates rather no change than gradual increase of the dose rate (Fig. 5). Because source-to-sample distance is the same in simulation and experiment, charge build-up as a function of grain size should also be the same. We discuss this in Sect. 5. The simulations also indicate that decreasing aliquot size enhances the dose rate by ∼ 10 %–20 % (Fig. 6). This significant gain of absorbed dose is probably caused by the secondary electron field and is discussed in Sect. 5.

## 4.4 Beta-source shape

There is compelling evidence from the experimental data (Figs. 3 and 4) that geometry and homogeneity of the irradiation field influence the dose rate. The effect of grain and

**Table 4.** Ratios between dose rates obtained from the three grain-size fractions and the three aliquot sizes used in the experiments. 180 : 90 is the ratio between the two coarse-grained samples; 4 : 90 and 4 : 180 are the ratios between the fine-grained and the coarse-grained samples. Errors are quoted at the 95 % confidence level resulting from the Student's $t$ distribution.

| Sample code | Grain-size ratio | Aliquot size 7.95 mm | Aliquot size 3 or 5 mm | Aliquot size 1 mm | Beta-source geometry |
|---|---|---|---|---|---|
| R17_180 : F19_90 | 180 : 90 | $1.005 \pm 0.004$ | $0.959 \pm 0.004$ | $1.010 \pm 0.002$ | open ring |
| | | $0.984 \pm 0.007$ | $1.001 \pm 0.006$ | $0.975 \pm 0.012$ | planar |
| R113_180 : F14_90 | | – | $0.862 \pm 0.008$ | $0.878 \pm 0.009$ | closed ring |
| R108_4 : F19_90 | 4 : 90 | $1.013 \pm 0.004$ | $1.019 \pm 0.003$ | $1.007 \pm 0.003$ | open ring |
| | | $0.939 \pm 0.006$ | $0.857 \pm 0.006$ | $0.869 \pm 0.006$ | planar |
| R108_4 : F14_90 | | – | $0.856 \pm 0.013$ | $0.794 \pm 0.013$ | closed ring |
| R108_4 : R17_180 | 4 : 180 | $1.018 \pm 0.003$ | $1.063 \pm 0.003$ | $0.996 \pm 0.003$ | open ring |
| | | $0.954 \pm 0.006$ | $0.857 \pm 0.006$ | $0.891 \pm 0.006$ | planar |
| R108_4 : R113_180 | | $0.979 \pm 0.013$ | $0.993 \pm 0.013$ | $0.905 \pm 0.013$ | closed ring |

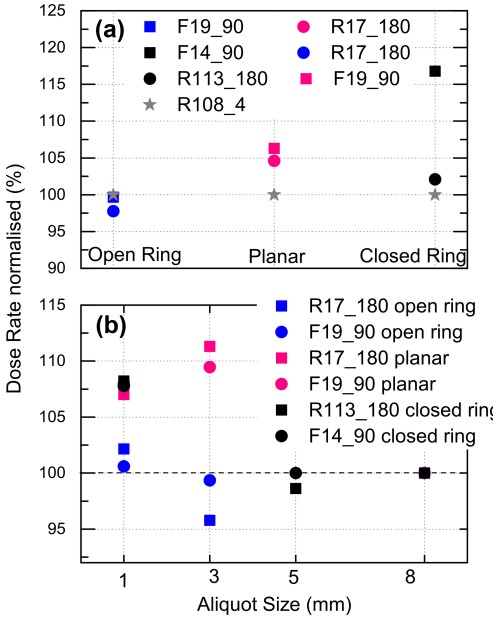

**Figure 3.** Experimentally determined normalised beta-dose rates. **(a)** Dose rate normalised to the respective fg value (sample R108_4) versus beta-source shape; aliquot size is 7.95 mm (R17_180; F19_90) or 5 mm (F14_90). **(b)** Dose rate normalised to 7.95 or 5 mm aliquot size plotted versus aliquot size. For the sake of clarity, error bars are not plotted. For data and uncertainty, see Table 3.

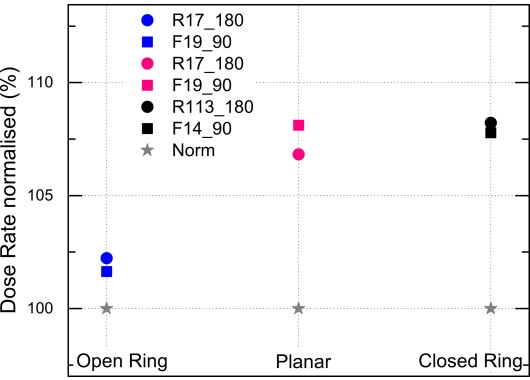

**Figure 4.** Beta-dose rates of 1 mm aliquots normalised to the 7.95 or 5 mm aliquot size of the respective sample versus beta-source shape. For the sake of clarity, error bars are not plotted. For data and uncertainty, see Table 3.

aliquot size is the smallest for the open-ring source due to its special design and is the biggest for the closed-ring source (Table 4). Because both sources simulated here (planar and closed ring) show the same response to aliquot size and grain size (Figs. 5, 6), we conclude that the shape of the source controls the magnitude of the dose rate. The generalised rule

seems to be correct in particular for large- and medium-sized aliquots but not for aliquot sizes < 5 mm (see details in Sect. 4.5). This is confirmed when simulating charge build-up as a function of depth in aliquot (Fig. 7): beyond the depth of approximately 150 mm, the magnitude of the build-up depends on aliquot size and source shape: the increase of dose rate is small in large aliquots irradiated by the closed ring source and significant in medium-to-large aliquots irradiated by the planar source. It is negligible in small aliquots regardless the shape of the beta source. For shallower depths (< 150 mm), the magnitude of build-up is enhanced by the electron backscatter of the ss cup (Fig. 7).

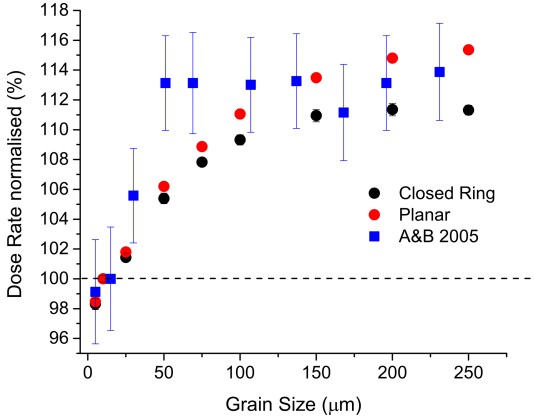

**Figure 5.** Result from *GEANT4* simulation compared to published experimental data. The dose rate is plotted as a function of grain size for the planar source and the closed-ring source and for experimental data (A&B 2005; Armitage and Bailey, 2005). Simulated data are normalised to the 10 µm grain size; aliquot size is 7.95 mm on stainless steel cup. Experimental data of A&B 2005 are normalised to the 15 µm grain size with aliquot size of 9 mm on aluminium disc.

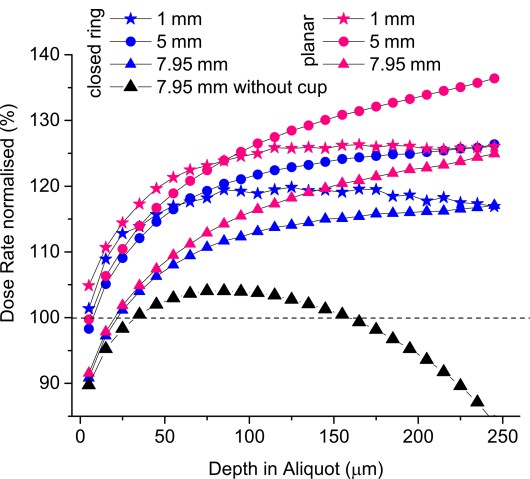

**Figure 7.** Result from *GEANT4* simulation: charge build-up in quartz grains of 250 µm size resting on a 7.95 mm ss cup compared to no cup as a function of depth in the sample for the two beta-source geometries. The sample is composed of 10 µm thick cylinder-shaped layers (see Fig. 1). Dose rate is normalised to the 10 µm layer and 7.95 mm aliquot size and represented in percent.

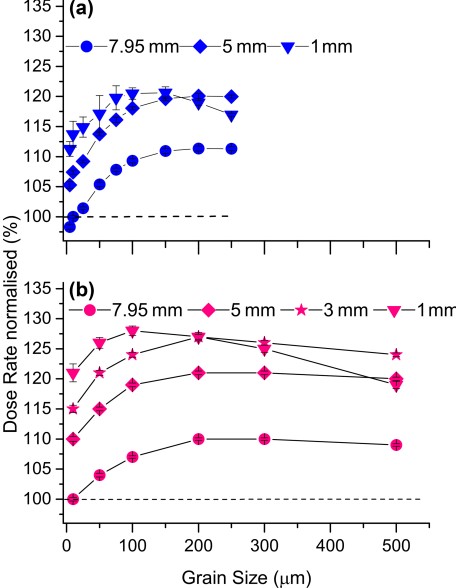

**Figure 6.** Result from simulations for dose rate as a function of grain size and aliquot size. Dose rate is normalised to the 10 µm grain size and 7.95 mm aliquot size expressed in percent; **(a)** *GEANT4* for the closed-ring beta source; **(b)** *MCNP6* for the planar beta source and grain sizes up to 500 µm to assess the significance of the trend.

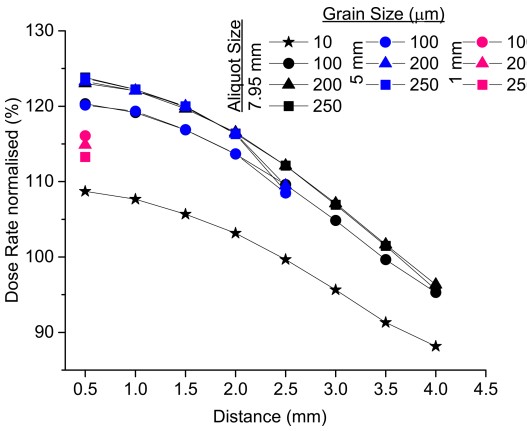

**Figure 8.** Result from *GEANT4* simulation: dose rate versus distance from centre of the stainless-steel cup for the closed-ring beta source. Data are for large (7.95 mm), medium (5 mm) and small (1 mm) aliquot sizes and for 10, 100, 200, and 250 µm grain sizes. Dose rate is normalised to 10 µm grain size, the average value of which is at 100 %.

## 4.5 Small aliquots

A drop of dose rate for grain sizes > 200 µm and aliquot sizes < 5 mm is evident from the dose deposition versus depth in grain (Fig. 7), from the comparison between grain and aliquot size (Fig. 6) and from the irradiation profile across the cup (Fig. 8). The experimental data show this drop only

for the planar source, albeit indistinguishably within uncertainties.

Beta particles interact with the aliquot and create secondary electrons that scatter around the interaction point. In the central part of the aliquot, the secondary particles interact with neighbouring grains or escape through the surface of the aliquot. If, however, the primary interaction occurs near the aliquot edge, the scattered electrons can also escape through the edge of the aliquot, not only through the surface. Therefore, the smaller the aliquot, the larger the percentage of escaping secondary electrons. Furthermore, the thicker the

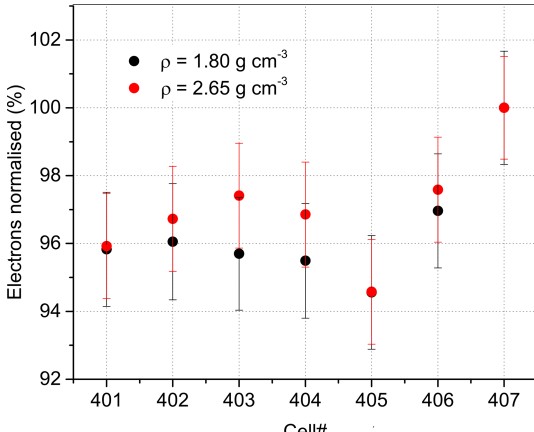

**Figure 9.** Result from *MCNP6* simulation: the number of electron-producing interactions plotted against cell number. The cell is a rounded $SiO_2$ grain of 300 μm diameter of the two densities displayed. Data are normalised to cell no. 407, which is the grain surrounded by other grains (for spatial arrangement of cells, see Fig. 2b).

aliquot, the smaller the percentage of secondary electrons escaping by the aliquot surface while the escape pathway via the edge remains the same. The edge effect is therefore governed by the ratio of grain size to aliquot size: the bigger the grain and the smaller the aliquot, the larger the reduction of the dose rate. In fact, the simulation shows that the number of scattered electrons decreases for the edge grains (Fig. 9). Thus, the edge effect counteracts the average increase of the beta-dose rate that occurs for decreasing aliquot sizes due to the radial increase of the dose rate towards the centre of the cup. It may even reverse if the ratio of grain size to aliquot size is appropriate, and the grains are located sufficiently far from the rim of the cup.

## 5  Discussion

The data presented here indicate that the dose rate of an individual beta source results from the interplay of a number of parameters. Most of these were identified by previous studies including grain-size-dependent build-up and attenuation of charge (e.g. Wintle and Aitken, 1977; Goedicke, 2007; Autzen et al., 2017). During SAR-based measurements using a $^{90}Sr / ^{90}Y$ beta source, incident beta particles penetrated the grain to a certain depth alongside backscattered electrons which had energies less than the initial source energy (Bell, 1980). Thus, the absorbed beta dose should decrease with increasing grain size (Wintle and Aitken, 1977; Goedicke, 2007; Hansen et al., 2018). That is why Hansen et al. (2018), building on findings of Greilich et al. (2008), attribute the undesirable overdispersion of their calibration value to variation of grain shape and volume because low-energy beta particles are increasingly attenuated in grains

> 100 μm, as already described by Bell (1980). In our simulation, however, charge build-up overcompensates the effects of attenuation resulting in a sustained rise of absorbed beta dose in grains > 150 μm resting on material of relatively high Z (Fig. 8). As a consequence, the simulation shows a continued rise of dose for grains 10–300 μm (Fig. 7) with a flattening of the rise above ∼ 150 μm grain size. This is arguably different but not too dissimilar to datasets deduced from experiments: Geodicke (2007) show an initial rise of dose up to 25–50 μm grain size, followed by a dose plateau for grain sizes 40–130 μm and a decrease for grains > 200 μm and Armitage and Bailey (2005) show a rise to ∼ 40 μm followed by a "jump" to a dose plateau for 50–250 μm grains. Thus, the competing mechanisms of build-up and attenuation lead to divergent dose-rate results mainly for ∼ 50–200 μm grains, likely caused by the geometry of the irradiation field (Bell, 1980).

Large aliquots show the expected build-up of charge with increasing grain size towards secondary equilibrium and small aliquots show the expected larger absorbed dose (Figs. S3–S6) due to the radial increase of dose rate towards the centre of the sample carrier (e.g. Spooner and Allsop, 2000; Veronese et al., 2007). This aliquot-size effect was indeed already highlighted in earlier studies (e.g. Bailiff, 1980; Bell, 1980). Our study provides evidence for further differentiating the aliquot-size effect: the dose enhancement in small aliquots is not the same in the simulation and experiment and is not the same for all grain sizes. The differences are caused by different penetration depths in grains and by the changing effect of backscattered electrons. The interplay seems to have the most variable effect on 50–100 μm grains mounted as aliquots of < 8 mm size (Fig. S8). The dose enhancement is likely reversed when large grains (i.e. > 200 μm) are mounted as small aliquots because with this geometry the probability of backscattered beta particles hitting the edge grain is reduced. However, this edge effect remains to be investigated in greater detail because with changing sphericity of grains (e.g. Autzen et al., 2017) and with potentially changing density of grain packing when the ideal grain monolayer is not achieved, the probability of beta interaction changes as well.

We also show that the shape of the beta source controls the magnitude of the absorbed dose and hence the build-up of charge. The fact that the dose absorbed in small grains must be lower than the dose absorbed in large grains is masked by the ring sources for which fine and coarse grains may absorb the same dose depending on the size of the aliquot (Fig. 4a). The open-ring source shows differences that are statistically negligible for all geometries, suggesting that homogeneity of the source associated with special design reduces the effect of grain and aliquot size on the calibration value.

Autzen et al. (2017) recommend minimising shape and volume variation of sample grains used for calibration, but our data suggest using multiple grain-size fractions for calibration. We think that as long as the sample originates from a

natural sedimentary deposit, either way it includes grains of various shape and form. We echo Goedicke (2007) in that the calibration procedure should employ small (4–20 μm), medium (20–80 μm), large (80–200 μm) and very large (200–300 μm) grain sizes. In addition, these grain-size fractions should be measured with small, medium and large aliquots. Calibrating all possible irradiation geometries of an individual beta source appears to be more important the more inhomogeneous a source is, and because source homogeneity is virtually unknown, the calibration procedure must take geometry into account. This will improve the accuracy of the calibration value with respect to the unknown natural sample.

Within the limits of the SAR protocol, the experimental uncertainty of the calibration value is usually reasonably low, thanks to the purpose-prepared sample material. However, with regard to beta-source calibration, a higher precision is desirable. Burbidge et al. (2016) show that parallel multiple-aliquot calibration transfer provides better accuracy and precision than single-aliquot measurements on single-dosed samples. Bos et al. (2006) show that the uncertainty can be reduced to 0.9 %. Their procedure envisages, first, a calculation of the administered gamma dose through Fricke solutions and, second, gamma irradiating several subsamples each with a different dose (e.g. 5, 10, ... 30 Gy). The determined beta $D_e$ values (s) are then plotted versus the gamma doses (Gy) and the inverse slope of the fitted line gives the beta-dose rate (Bos et al., 2006). The total uncertainty is derived from the uncertainties of beta and gamma irradiation. We therefore say that the laboratory's key parameter can be improved in terms of accuracy and precision by including several grain sizes, several aliquot sizes and several gamma doses in the calibration experiments.

## 6 Conclusion

With the number of parameters in mind, it is clear that predicting the dose rate through a series of simulations is too laborious in comparison to a series of relatively simple SAR-based experiments. Here, indeed additional work is required to better estimate the impact of the edge effect on dose rate. If the experimental approach is the way forward, then effort should be made to improve accuracy and precision of the calibration value. Future work should therefore focus on gamma irradiating a calibration sample of several grain-size ranges with several gamma doses in order to determine the value from the regression line and not from a single data point.

**Code availability.** The Geant4 code developed for simulating beta irradiation by the Lexsyg SMART planar source and the Lexsyg SMART ring source is based on Geant4 version 10.1.2 and available in the Supplement. It includes the [90]Sr beta spectra used for the simulations. Geant4 version 10.1.2 is available under the Geant4 licence at http://cern.ch/geant4/license (last access: 14 June

2021). Please contact Loic Martin at loic.martin@glasgow.ac.uk for technical questions. The Monte Carlo radiation transport code (MCNP6.2) is commercial and licenced.

**Data availability.** Data required to create input files for the MCNP6.2 code are available through references quoted in the text or in the Supplement. Data for the geometry of the beta sources of the Lexsyg luminescence readers of Freiberg Instruments are not publicly accessible and may be requested by the manufacturer.

**Supplement.** The supplement related to this article is available online at: https://doi.org/10.5194/gchron-3-1-2021-supplement.

**Author contributions.** BM and MD designed the experiments, and CB and CT carried them out. LM and MD performed the simulations. BM, LM, MD, NM, SK and AL discussed details of results. BM prepared the manuscript with contributions from all co-authors.

**Competing interests.** The authors declare that they have no conflict of interest.

**Acknowledgements.** We wish to thank Andreas Richter (Freiberg Instruments) for his helpful advice regarding the design of the beta sources of the Lexsyg instruments. Sebastian Kreutzer received funding from the European Union's Horizon 2020 research and innovation programme (see below). We would like to thank two anonymous reviewers for their constructive and helpful comments.

**Financial support.** This research has been supported by the European Union's Horizon 2020 research and innovation programme under the Marie Skłodowska-Curie grant agreement no. 844457 (CREDit) and the Conseil Regional Nouvelle Aquitaine (project DAPRES_LA_FEM).

**Review statement.** This paper was edited by Julie Durcan and reviewed by two anonymous referees.

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

## Remarks from the typesetter