# Peer review of "Technical note: On the reliability of laboratory beta-source calibration for luminescence dating"

_Geochronology, 2020_

## Referee Comment (RC1) · Anonymous Referee #1 · 16 Dec 2020

The manuscript presents new data on systematic effects encountered with regard to beta source calibration in luminescence dating. As the authors claim, source calibration is a central part in obtaining reliable luminescence dating results and systematic errors should be avoided whenever possible. While in the past 40 years numerous papers on that topic were published, the authors draw particular attention to the aliquot size and its impact on the calibration results, an aspect that has not been considered previously.

The dataset compiled by the authors on three different luminescence readers with various source geometries, grain sizes and aliquot sizes supports the conclusions drawn, as well as do the results of GEANT4 modelling in general. There are some slight discrepancies between experimental and modelled data that remain unexplained though. The same is true for the "edge effect" that the authors think to be the reason for the

observed dose rate dependence on aliquot size, but any details on underlying mechanisms are not provided. However, given the scope of the submission ("technical note") it would probably go too far to ask the authors for more in-depth analyses of related physical processes. In that sense, the study represents a valuable contribution for all luminescence practitioners and hopefully entails an enduring effect on how the sources will be calibrated in luminescence laboratories. I recommend publication after the issues detailed below have been taken into account.

Another aspect of the paper, which is rather mentioned on the fly, is the use of several gamma doses for beta source calibration and then taking the regression of a plot of gamma dose vs. recovered beta dose to derive the dose rate of the beta source. This is an approach that should certainly be promoted, and hence deserves a bit more weight in the manuscript (currently it is mainly mentioned at the end of the discussion), although the idea is not new (Bos et al., 2006, as correctly stated by the authors).

The manuscript is well written and logically structured, although some aspects of data presentation should be improved, as outlined below. I found some formal inconsistencies and oversights that need streamlining, such as the use of the terms "beta source" vs "beta-source" or the erroneous use of the unit "mm" instead of "$\mu$m". This can be easily done with a thorough round of proof-reading. Also, it seems that a wrong plot has been uploaded as Fig. 4; the authors should check this carefully. Some of the references in the bibliography are incomplete and should be checked also.

Specific comments

l. 22: What do you mean with "geometrical function"? Maybe just replace by "irradiation geometry"?

l. 39: What is meant with the "interplay between sample and sample carrier"? Please be a bit more specific here.

l. 75 (Table 1 caption): it should read "... are derived from MC simulation".

[Figure]

l. 79: I feel that the calibration quartz from Freiberg needs to be described regarding its main properties (origin, preparation, treatment before giving the gamma dose). While readers can look up in Hansen et al. (2015) for the DTU quartz, nothing is written here about the Freiberg quartz.

l. 87 (Fig. 1): The font size for most of the information contained in the figure is too small. On a print-out the letters can hardly be read. Please increase the font size.

l. 99 (Table 2 caption): Which stimulation power density was used for which aliquot size and why? Please provide the reasoning why you chose this approach.

Table 2, last row: it should be "R108_4", I guess.

l. 108: Does "depth of dose rate" mean "dose rate as a function of depth"? Consider re-phrasing.

l. 109: The units given for the simulated layers of the sample "cylinder" should be $\mu$m instead of mm, I would think (same in the caption to Fig. 2).

l. 130: Does this statement in brackets mean that the dose is registered in Gy for each starting particle, i.e. particle emitted from the source? In general, for readers not so familiar with simulation of irradiation, it would be beneficial to write one or two additional sentences on the specific purposes of the GEANT4 and MCNP6 codes, i.e. which code was used for which part of the simulation. In the current version of the text, this is not very obvious.

Table 3: Are the uncertainties for the dose rate given at 1-sigma CI (68%)? If yes, please add this information. It would further add the immediate comprehension of the table if an additional column with the "sample code" would be added. This information is provided in Table 2, but if given here as well correlations between the calibration results and the types of calibration samples could be established much easier. Finally, please homogenise the number of significant digits for the reported dose rates.

l. 157: How does the "total uncertainty of experimental data" of 5-8% relate to the

dose rate results shown in Table 3 (where the 1-sigma (?) uncertainty is in the range 2.0-3.4%)?

l. 165: It should read "... quoted at 95% confidence level...".

Table 4: I suggest expressing the difference in dose rate between different grain size fractions not in percent, but as a ratio, because in this way the direction of deviation is also indicated (similar to Table 3).

Fig. 3: Please mind the consistency of axis labelling (Dose Rate vs. Dose rate).

Fig. 4: This figure seems to be identical with Fig. 3a. Please check and update.

Fig. 5: Black and red dots in the caption (legend to plot) should be swapped.

Fig. 6: Why are the GEANT4 simulations skipped for grain sizes >250 $\mu$m, while they were carried out for the MCNP6 simulation?

Fig. 7: What is the purpose and meaning of the simulation without sample holder? Can inferences be made about the role/magnitude of electron backscatter from the sample holder? Maybe this aspect should be shortly discussed in the manuscript.

Fig. 8: If the dose rate is shown normalized to the 10 $\mu$m large aliquot simulations, why do the data start at ∼108% in the center of the sample carrier?

l. 236: It should read "linearly".

l. 234-242: Even after reading the paragraph several times, I must admit that I do not fully understand the logic behind it. Maybe it should be mentioned that the surface of a grain is approximated here as a 2D spherical plane, and not as understood as the real surface of a 3D spherical grain (if I got it right...). Probably this paragraph needs re-phrasing to clarify what you mean with the "edge effect". This affects also the discussion (l. 270-274), which should be adapted/expanded to add some explanations on the edge effect. Additionally, in line 239, it should be a reduction of absorbed dose by ∼3-5% (as compared to Fig. 8).

[Figure]

l. 249: ". . . of an individual beta source results from . . ."

l. 265: Consider writing "geometry" instead of "geometrical function".

l. 278: Figure reference correct? There is no Fig. 4A (and Fig. 4 is probably the wrong one. . .).

l. 288: What is "purpose-prepared sample material"?

l. 293: Replace "Des" by "De values".

---

## Author Comment (AC1) · 21 Dec 2020

We wish to thank the reviewer for their constructive and detailed comments. We will take all points into account when revising the manuscript.

---

## Referee Comment (RC2) · Anonymous Referee #2 · 7 Jan 2021

General comments

This technical note presents the results of an investigation into accurate calibration of beta sources for use in luminescence dosimetry. It presents experimental and modelled data, using these to explore dose rate variations caused by grain size, aliquot diameter and source geometry. The experiments are performed using lexsyg instruments, and some of the results (notably those relating to different source geometries) will be specific to these instruments. Observations regarding grain size and aliquot diameter are likely to be more generally applicable. Beta source calibration is an important issue in luminescence dosimetry, and a number of recent publications have demonstrated that the production of accurate calibrations is not trivial. This technical note has the potential to make an important contribution to that literature. However, the
manuscript contains a large number of loose ends, errors and unexplained assumptions. Individually none of these points are critical, but they are numerous, meaning that the manuscript will require considerable "tidying up" prior to publication.

Specific comments

1) Throughout the manuscript the authors imply that variation in dose rate due to variation in aliquot diameter is a novel observation. This isn't true since Spooner and Allsop (2000) investigated the spatial variation of dose rate from a number of 90Sr/90Y sources, concluding that there was a measurable decrease with distance from the centre of the sample holder. They provided a table (their Table 1) listing dose-rate correction factors for different aliquot diameters, and cited two previous studies (Zimmerman, 1970; Spooner, 1987) yielding similar results. The present paper extends these studies, but the concept of a change in dose rate from the centre to the periphery of a sample holder is not in itself new.

2) When considering the experimental data regarding dose rate variation with aliquot diameter, the authors appear to assume that the activity across the face of the source is homogenous. If this isn't the case, and the literature contains a number of studies demonstrating the existence of inhomogenous sources (e.g. Ballarini et al., 2006; Pawlyta et al., 2019), then effects attributed to aliquot diameter or irradiation geometry may actually result from non-uniform distribution of radioisotopes across the face of the source. In the present study, data were produced using only one example of each source type, meaning that it is impossible to distinguish between source inhomogeneity and other effects. As a minimum, the authors should acknowledge this potential limitation, though the best solution would be to test the homogeneity of the sources directly (e.g. Pawlyta et al., 2019) or make the same measurements on multiple examples of each source type to determine whether the pattern of change remains constant.

3) Dose rate homogeneity across the face of the sample holder (Spooner and Allsop, 2000) and the variation of dose rate with grain size (e.g. Armitage and Bailey, 2006)
[Figure]

are both through to vary with source-to-sample spacing, and possibly also with the dimensions of the active face of the beta source (Spooner and Allsop, 2000). It would be very helpful to the reader if this information was given for each instrument and source used in the present study, since it would allow more meaningful interrogation of the data presented. For example, does the "open ring" have a larger active face relative to the size of the sample holder than for other sources? If so, this might explain why aliquot size does not appear to be an important consideration for this source.

4) The number of aliquots used to produce each datapoint in the experimental datasets is often rather low. For example, only two 8 mm diameter aliquots of R108_4 were measured using the lexsyg SMART closed ring source, and yet all the data for that source in Figure 3 are normalised to that point. Is this really sensible? Well over half the experimental datasets are constructed from five aliquots or fewer. I accept that the calibration samples used are bright and highly reproducible, but reliance on very small numbers of aliquots makes the resulting dataset prone to other sources of uncertainty e.g. non-uniform aliquot preparation, grain(s) between the sample and hotplate, deformed sample holders (Duller et al., 2000) etc etc. The authors should at least comment on why they think that the small number of aliquots measured doesn't pose a problem.

5) There are a large number of errors in the figures and tables. a) Three aliquot diameters were measured for sample R113_180 using the lexsyg SMART closed ring source, yet only data for the 5 mm diameter aliquots are provided in Table 2. b) In Figure 3a, the caption states that all data are for the 8 mm diameter aliquots, yet this size wasn't measured for sample F14_90 using the lexsyg SMART closed ring source, and the data presented appear to be for the 5 mm aliquots. Similarly, the data in Figure 3b are normalised to the 8 mm diameter aliquots, but I suspect that the 5 mm datapoint for F14_90 closed ring source is the 5 mm data normalised to itself, which is misleading. c) Figure 4 is the same as Figure 3A – I suspect that Figure S1 was actually intended to be Figure 4, in which case what is Figure S1? Whatever the answer, the correct

Figure 4 needs to be inserted. d) The data in Figure 7 claim to be normalised to the 10 m grain size and 7.95 mm (reported as 8 mm elsewhere in the paper) aliquot diameter, whereas I suspect they are normalised to the 10 m, 5 mm aliquot. Also, why (and how?) did you perform the GEANT4 simulation without the sample cup? e) I suspect that the data in Figure 8 are normalised to the 10 $\mu$m 2.5 mm point in the 7.95 mm dataset, though the caption states that data are normalised to the "10 $\mu$m grain size and large aliquot", which I think is the entire starred dataset. Please clarify.

6) I don't really understand the paragraph starting on line 234, which is problematic because it appears to be critical to your explanation of grain size effects with small aliquots. Please expand and clarify.

Technical points

Line 65: The phrase "ring shaped source closed to the top" is awkward. As I understand it there is an "open ring" where the active face is a donut shape constructed of 17 "mini-sources" whereas the "closed ring" is an approximately circular active face constructed from 23 "mini sources". Diagrams in the SI would help clarify this point and are probably required to answer point 3 above anyway. Then, the terminology "open" and "closed" ring would be easier to understand.

Figure 1. Please make the small text larger.

Line 99: Please explain the rationale for using different stimulation powers for different sized aliquots and show which power was used for which size.

Line 109: Here and elsewhere I think the sample cylinder layers are 5 or 10 $\mu$m thick not 5 or 10 mm.

Line 110: Why was a density of 1.8 g/cm3 used for the SiO2? This is a good approximation for the density of a homogenous quartz sand with no matrix i.e. an idealised natural sediment from which a luminescence sample could be taken. However, if I understand the modelling work correctly, the sample is represented as stacked discs

of pure SiO2 i.e. it should have a 2.65 g/cm3. If the reduced density is a modeller's approximation of sand deposited as a monolayer on a sample holder, please explain the logic – I won't be the only non-modeller to read this paper.

Figure 2: Please add A and B to the figures. Please also provide a clearer explanation of exactly what Figure 2B is simulating – possibly your revision of the paragraph beginning on line 234 will help with this explanation (see point 6 above). Caption line 119 "plan-view".

Line 153: Here and throughout the paper the terms "fine" and "coarse" grains are used. I accept that this is standard laboratory slang for different grain sizes, but it would be better to be specific e.g. 4-11 $\mu$m, particularly given that you use two different sizes of "coarse grain" e.g. line 164. Even if you don't make this change, please move the "(fg=fine grain)" statement somewhere else since it is currently out of place.

Table 3 and 4: Dashed and solid horizontal lines appear more or less at random. Please standardise.

Line 179: You state that for 8 mm aliquots the effect of grain size is insignificant. By what criterion? If by your own stated in Line 171, please say so. I assume that this is the case since the percentage difference for the "4:90" ratio is 6.144$\pm$0.006 i.e. a ratio of 1.06144$\pm$0.00006, which must be statistically significantly different from unity?

Line 181: "6-26%" should read "0.6-26%".

Line 182: "...the magnitude of the difference is controlled by the shape of the source". This sentence requires more explanation.

Sentence starting on Line 182: Assuming this sentence is actually referring to Figure S1, this statement isn't true for the 180-250 $\mu$m data measured using the closed ring source.

Line 192: I'm not convinced that the Armitage and Bailey (2006) data "jump" between 50 and 100 $\mu$m – it appears that the dose rate increases fairly linearly with grain size

from 0-50 $\mu$m, after which it plateaus. As the authors suggest (line 191), the experimental dataset is actually rather similar to the simulated data.

Line 224: I think this should read ">5%" not "<5%".

Line 239: The figure reference should probably be to Fig. 9, in which case a reduction in the absorbed dose of $\sim$3% looks more appropriate by eye.

---

## Author Comment (AC2) · 4 Feb 2021

We appreciate your review – it is clear and detailed and will improve our paper, so thank you for that. We are very sorry for the mistake with Fig. 4.

*There are some slight discrepancies between experimental and modelled data that remain unexplained though*
We agree – the edge effect is the most obvious one. We will add a figure in the supplement which illustrates the relationship between grain, aliquot and cup with respect to incident and escaping beta particles. The text (sec 4.5) will say:

Beta particles interact with the aliquot and create secondary electrons that scatter around the interaction point. In the central part of the aliquot the secondary particles interact with neighbouring grains or escape by the surface of the aliquot. If, however, the primary interaction occurs near the aliquot edge, the scattered electrons can, in addition, escape through the edge of the aliquot. The smaller the aliquot, the larger the percentage of escaping secondary electrons. Furthermore, the thicker the aliquot, the smaller the percentage of secondary electrons escaping by the aliquot surface while the escape pathway via the edge remains the same. The edge effect is therefore governed by the ratio grain size to aliquot size: the bigger the grain and the smaller the aliquot the larger the reduction of the dose-rate. In fact, the simulation shows that the number of scattered electrons decreases for the edge grains (Fig. 9). Thus, the edge effect counteracts the average beta-dose rate increase that occurs for decreasing aliquot sizes because of the radial increase of dose rate towards the centre of the cup. Possibly, it even reverses it when the ratio of grain size to aliquot size is appropriate and the grains are situated sufficiently away from the rim of the cup.

In addition, we should have stated more clearly that with given source-probe distance the radiation field is curved implying a radial decrease of dose rate from the centre to the periphery as outlined by several studies (e.g., Spooner and Allsop, 2000). This will be added to sec 2.1 (where sources are described) by saying: With a source sample distance of 6.9 mm the radiation field of all sources is expected to be curved. Veronese et al. (2007) describe the curve with a power function yielding a parabolic curve of variable width. A very wide, hence relatively flat, parabolic curve is delivered by the open-ring source (Richter et al., 2012) due to its special design.

*the use of several gamma doses for beta source calibration and then taking the regression of a plot of gamma dose vs. recovered beta dose to derive the dose rate of the beta source. This is an approach that should certainly be promoted, and hence deserves a bit more weight in the manuscript*
We agree. We will emphasise the approach in the abstract by saying:

We conclude that future calibration samples should consist of subsamples composed of small, medium, large and very large quartz grains each obtaining several gamma doses. The calibration value measured with small, medium and large aliquots is then obtained from the inverse slope of the fitted line, not from a single data point. In this way all possible irradiation geometries of an individual beta source are covered, and the precision of the calibration is improved.

*Why are the GEANT4 simulations skipped for grain sizes >250 _m, while they were carried out for the MCNP6 simulation? -* This was simply motivated by workload of the respective expert and/or the demand on the computational resource.

*Fig. 4: This figure seems to be identical with Fig. 3a. Please check and update.*
Apologies for this mistake. Here it is.

[Figure]

Fig. 4. Beta-dose rates of 1 mm aliquots normalised to the 8 mm aliquot size of the respective sample versus beta-source shape. For data and uncertainty see Table 3.

*Fig. 7 - What is the purpose and meaning of the simulation without sample holder? Can inferences be made about the role/magnitude of electron backscatter from the sampleholder? Maybe this aspect should be shortly discussed in the manuscript.*

Indeed, the purpose was to show the magnitude of electron backscatter without sample carrier. It should have been discussed in the text and we apologies for this oversight. The discussion will be included in sec 4.4 where it says "This [shape of source] is confirmed when simulating charge build-up as a function of depth in aliquot (Fig. 7). Beyond the depth of ca 150 mm the magnitude of the build-up depends on aliquot size and source shape: the increase of dose rate is small in large aliquots irradiated by the closed ring source and significant in medium to large aliquots irradiated by the planar source. It is negligible in small aliquots regardless the shape of the beta-source. For shallower depths (<150 mm) the magnitude of build-up is enhanced by the electron backscatter of the ss-cup (Fig. 7).

*Fig. 8: If the dose rate is shown normalized to the 10 _m large aliquot simulations, why do the data start at _108% in the center of the sample carrier?*

The figure shows the dose rate profile for various grain- and aliquot sizes. It does not show the average dose rate. The normalised average value of the 10 μm curve is at 100%. We will clarify this is the figure caption.

*What is "purpose-prepared sample material"?* – it is a natural sample (e.g. dune sand) prepared for the purpose of becoming beta-source calibration material. Here it is annealing and repeated irradiation and read-out using blue-light stimulation in order to sensitise the quartz.

---

## Author Comment (AC3) · 4 Feb 2021

We thank the referee for their detailed review and reply as follows.

*Throughout the manuscript the authors imply that variation in dose rate due to variation in aliquot diameter is a novel observation. This isn't true since Spooner and Allsop (2000) investigated… The present paper extends these studies, but the concept of a change in dose rate from the centre to the periphery of a sample holder is not in itself new*

The referee makes a good point by highlighting the importance of irradiation field uniformity when geometrical differences of dose rate are detected. We were aware of this issue before starting the experiments and stated this in the text (line 66ff). The sentence may well be insufficient but makes clear that we do not pretend to come up with a new concept on radial dose-rate change. Instead, we present a study which extends the 'size matters' concept (e.g., Duller, 2008) and builds on the conclusion of many studies, ever since single-grain dating is an option: non-uniformity must be considered when calibrating the source (e.g., Ballarini et al.2 006, Veronese et al. 2007). While we hope that this statement cancels the claim of the reviewer, we acknowledge the need for saying more about source uniformity in the text. In sec 2.1 we will describe the sources in terms of (unknown) homogeneity and shape of the radiation field by saying

The ring-shaped sources consist of mini-sources which were tested for homogeneous activity (<5% variation) for the open-ring source before mounting in a circular groove (14 mm diameter) of the stainless-steel housing (Richter et al., 2012). The radiation field of this source varies by 2-8% across 8-10 mm cup diameter (Richter et al., 2012). The larger variation occurs towards the cup edge due to increasing backscatter from the cup rim (Fig. 1), but the inner 6 mm of the cup is exposed to a very homogeneous radiation field (Richter et al., 2012). The sources of the *SMART* readers are not pre-selected for homogeneous activity and may deliver a less uniform radiation field. With a source sample distance of 6.9 mm the radiation field of all sources is expected to be curved. Veronese et al. (2007) show that the dose-rate reduction follows a power function which yields a parabolic curve of variable width. A very wide, hence flat parabolic curve is delivered by the open-ring source (Richter et al., 2012) due to its special design. Before starting the experiments, the beta sources in the readers were manually adjusted to align the centres of sample carrier and source aperture (Discher et al., 2021). Thereby, an almost symmetrical irradiation field across the cup was achieved where the width of the parabola depends on the design of the source.

*When considering the experimental data regarding dose rate variation with aliquot diameter, the authors appear to assume that the activity across the face of the source is homogenous. If this isn't the case, and the literature contains a number of studies demonstrating the existence of inhomogenous sources (e.g. Ballarini et al., 2006; Pawlyta et al., 2019), then effects attributed to aliquot diameter or irradiation geometry may actually result from non-uniform distribution of radioisotopes across the face of the source.*

We are sorry that the referee came to this impression. We do not make assumptions on the homogeneity of the sources used for the experiments other than that they provide a bell-shaped irradiation field and this is confirmed by the simulation (Fig. 8). Our intention was to show that sources should be calibrated for every irradiation geometry used in dating application. Even if inhomogeneity is quantified using a scintillator or similar tools, it would only confirm the take-home message of our technical note: calibrate every possible irradiation geometry that you use in dating

application because the inhomogeneity itself cannot be addressed by the practitioner. Furthermore, our simulation was conducted on the basis of a homogeneous source, i.e., radioisotopes are evenly distributed across the source face. Details of the simulation results displayed in Figs S3-S6 and summarised in Figs S7 and S8 show that there is a dose enhancement as aliquot size decreases. No doubt, there is more to do to unravel the complex interaction, but we feel we have shown enough data that support our take-home message.

*…data were produced using only one example of each source type, meaning that it is impossible to distinguish between source inhomogeneity and other effects.*
We feel this statement misses the aim of our study which was clearly outlined in the introduction (unexplained overdispersion of calibration data, etc). Our aim was not to study variations of individual source types. That said, we endorse the statement: our experimental data are not good enough to differentiate between source inhomogeneity and other effects because the 95% CL uncertainties do not allow to disentangle the interplay of various parameters. Here again, the argument the simulation results should satisfy the critique: in the simulation beta particles emitted by the source were evenly distributed – for details see our previous answer.

*As a minimum, the authors should acknowledge this potential limitation, though the best solution would be to test the homogeneity of the sources directly (e.g. Pawlyta et al., 2019) or make the same measurements on multiple examples of each source type to determine whether the pattern of change remains constant*
Yes indeed, only one example of each source type was used for the experiments and only two source types (planar and closed-ring) were simulated. The objective of the study was to identify the impact of some parameters on the calibration value, we did not intent to study how variable a particular source type is as a result of the manufacturing process. Given that source (in-)homogeneity is an unchangeable fact for the practitioner, Pawlyta's scintillator method remains useful, but does not supersede our calibration approach. Moreover, the scintillator method seems to require direct contact between humans and source (when inserting it in a specially designed socket), hence radiation protection permission and specialised software for data analysis – all together not really readily available in most luminescence laboratory, whereas our recommendation can be implemented straightaway by every practitioner.

*Dose rate homogeneity across the face of the sample holder (Spooner and Allsop, 2000) and the variation of dose rate with grain size (e.g. Armitage and Bailey, 2006) are both through to vary with source-to-sample spacing, and possibly also with the dimensions of the active face of the beta source (Spooner and Allsop, 2000).*
We think this statement mixes a number of different parameters in a way that is misleading. Dose-rate uniformity across the face of the sample holder does not really exist, at least not for the source-probe distances typically used in luminescence readers for which the radiation field is curved. According to Veronese et al. (2007) this curve takes the form of a parabola. The dose rate changes with changing distance, but for the source-probe distances typically used in luminescence readers

the impact on the shape of radiation field is negligible (Spooner and Allsop, 2000; Veronese et al., 2007). The dimension of the active face is as important as the source aperture is, the design of the source housing and its material. The details are described in the references relevant for each source (e.g. Richter et al. 2012).

*It would be very helpful to the reader if this information was given for each instrument and source used in the present study, since it would allow more meaningful interrogation of the data presented. For example, does the "open ring" have a larger active face relative to the size of the sample holder than for other sources? If so, this might explain why aliquot size does not appear to be an important consideration for this source.*

Thank you for indicating a weakness in the manuscript. First, for the open-ring source we should have cited Richter et al. 2012 (ATL) - apologies for providing the wrong reference. Second, we should have highlighted the radiation field of this source in comparison to the two other sources. We will add this in sec 2.1 as stated above.

*The number of aliquots used to produce each datapoint in the experimental datasets is often rather low. For example, only two 8 mm diameter aliquots of R108_4 were measured using the lexsyg SMART closed ring source….. I accept that the calibration samples used are bright and highly reproducible, but reliance on very small numbers of aliquots makes the resulting dataset prone to other sources of uncertainty e.g. non-uniform aliquot preparation, grain(s) between the sample and hotplate, deformed sample holders (Duller et al., 2000) etc etc. The authors should at least comment on why they think that the small number of aliquots measured doesn't pose a problem.*

The number of measured aliquots is indeed small, but, as the referee states correctly, the material is "highly reproducible". If errors are assumed to be independent and normally distributed, the small number of aliquots is not an issue as long as the uncertainties are calculated using the student's t distribution. The values with their uncertainties are then representative of the measured effects and comparable to any other data set. We will re-calculate the uncertainties. It should be said at this point that total uncertainties include the applied gamma-dose uncertainty (see Tribolo et al. 2019 for calculation). However the experimental uncertainties are calculated, for the purpose of our study they are too big to work out the 'size matters' calibration issue, hence the need for simulations.

*Errors in Figs and Tables*

We thank the reviewer for helping with tidying up the manuscript. We will iron out the issues.

*why (and how?) did you perform the GEANT4 simulation without the sample cup?*

The purpose was to show the magnitude of electron backscatter without sample. It should have been discussed in the text and we apologies for this oversight. We will include this in sec 4.4 where it says: This [shape of source] is confirmed when simulating charge build-up as a function of depth in aliquot (Fig. 7). Beyond the depth of ca 150 mm the magnitude of the build-up depends on aliquot size and source shape: the increase of dose rate is small in large aliquots irradiated by the closed ring source and significant in medium to large aliquots irradiated by the planar source. It is negligible in small aliquots regardless the beta-source shape. For shallower depths (<150 mm) the magnitude of build-up is enhanced by the electron backscatter of the ss-cup (Fig. 7).

*don't really understand the paragraph starting on line 234, which is problematic because it appears to be critical to your explanation of grain size effects with small aliquots. Please expand and clarify.*

We will add a figure in the supplement which illustrates the relationship between grain, aliquot and cup with respect to incident and escaping beta particles (Fig. S9). The text (sec 4.5) will say:

Beta particles interact with the aliquot and create secondary electrons that scatter around the interaction point. In the central part of the aliquot the secondary particles interact with neighbouring grains or escape by the surface of the aliquot. If, however, the primary interaction occurs near the aliquot edge, the scattered electrons can, in addition, escape through the edge of the aliquot. The smaller the aliquot, the bigger the percentage of escaping secondary electrons. Furthermore, the thicker the aliquot, the smaller the percentage of secondary electrons escaping by the aliquot surface while the escape pathway via the edge remains the same. The edge effect is therefore governed by the ratio grain size to aliquot size: the bigger the grain and the smaller the aliquot the larger the dose-rate loss. In fact, the simulation shows that the number of scattered electrons decreases for the edge grains (Fig. 9). Thus, the edge effect counteracts the average beta-dose rate increase that occurs for decreasing aliquot sizes because of the radial increase of dose rate towards the centre of the cup, Possibly, it even reverses it when the ratio of grain size to aliquot size is appropriate and the grains are situated sufficiently away from the rim of the cup.

*Line 65: The phrase "ring shaped source closed to the top" is awkward. As I understand it there is an "open ring" where the active face is a donut shape constructed of 17 "minisources" whereas the "closed ring" is an approximately circular active face constructed from 23 "mini sources". Diagrams in the SI would help clarify this point and are probably required to answer point 3 above anyway.*

We understand the desire to see the design of the sources tested here. For every source a reference is provided where the design is illustrated in great detail. We feel there is no need to copy these published figures into our manuscript. As stated above, the dimension of the source aperture matches the size of the sample carrier.

*Line 99: Please explain the rationale for using different stimulation powers for different sized aliquots and show which power was used for which size.*

The stimulation power of the blue LEDs (458±5 nm) was reduced with increase of aliquot size to avoid over-exposure of the photomultiplier. A sentence about over-exposure will be added.

---

## Author Response (AR1)

**Point-by-point reply to reviewers**

**Reviewer 1**

*the use of several gamma doses for beta source calibration and then taking the regression of a plot of gamma dose vs. recovered beta dose to derive the dose rate of the beta source. This is an approach that should certainly be promoted, and hence deserves a bit more weight in the manuscript* now added in the abstract

*l. 22: What do you mean with "geometrical function"? Maybe just replace by "irradiation geometry"?* Done

*l. 39: What is meant with the "interplay between sample and sample carrier"? Please be a bit more specific here.* Replaced by *"…*on the atomic numbers ($Z$) of mineral and sample carrier (line 40)

*l. 75 (Table 1 caption): it should read "∷ are derived from MC simulation".* Done

*While readers can look up in Hansen et al. (2015) for the DTU quartz, nothing is written here about the Freiberg quartz.* The Freiberg quartz is now published, and the reference is provided in line 83.

*Please increase the font size in Fig 1 –* done.

*Which stimulation power density was used for which aliquot size and why? Please provide the reasoning why you chose this approach?* The revised text says that the stimulation power was changed to avoid overexposure of the photomultiplier …depending on the size of the aliquot (header of table 2)

*Table 2, last row: it should be "R108_4" –* yes, done

*Does "depth of dose rate" mean "dose rate as a function of depth"? Consider re-phrasing.* Re-worded following the suggestion (line 120)

*The units given for the simulated layers of the sample "cylinder" should be _m instead of mm, I would think (same in the caption to Fig. 2).* Yes, and sorry for this formatting error.

*l. 130: Does this statement in brackets mean that the dose is registered in Gy for each starting particle, i.e. particle emitted from the source?* In the source code the results of the F6 tally are reported in dose per source particle (Gy) averaged over the target cell. Because MCNP results are normalised in our study the unit of the F6 tally is not relevant, we have removed the unit in the text (line 141). In the supplement 'number of tracks' are reported – these are a direct output of the code.

*write one or two additional sentences on the specific purposes of the GEANT4 and MCNP6 codes, i.e. which code was used for which part of the simulation.* Done (lines120 and 140)

*Improve Table 3: sample code, number of digits.* Done

*Line 157 - How does the "total uncertainty of experimental data" of 5-8% relate…*1 and 2 sigma errors were used for comparing data, but because *n* (number of aliquots measured) is low, these are actually not statistical errors suitable for comparison. We have re-calculated all errors and listed

these in Table 3. Obviously, errors are big and would only become small when increasing *n*. On the other hand, the accuracy of the value will not change with *n*>20 due to the excellent reproducibility of the calibration quartz. Thus, error bars are not plotted in Figs 3 and 4.

*It should read ": : : quoted at…* Done

*Table 4: I suggest expressing the difference in dose rate between different grain size fractions not in percent, but as a ratio* - done

*consistency of axis labelling* – thanks yes, done.

*Why are the GEANT4 simulations skipped for grain sizes >250 _m, while they were carried out for the MCNP6 simulation?* - This was simply motivated by workload of the respective expert and/or the demand on the computational resource.

*Fig. 4: This figure seems to be identical with Fig. 3a. Please check and update.* Done

*Black and red dots in the caption (legend to plot) should be swapped.* Done

*Fig. 7 - What is the purpose and meaning of the simulation without sample holder? Can inferences be made about the role/magnitude of electron backscatter from the sampleholder? Maybe this aspect should be shortly discussed in the manuscript.* Explanation now added (lines 230)

*Fig. 8: If the dose rate is shown normalized to the 10 _m large aliquot simulations, why do the data start at _108% in the center of the sample carrier?* The figure shows the dose rate profile for various grain- and aliquot sizes. It does not show the average dose rate. The normalised average value of the 10 μm curve is at 100%. Now explained in the figure caption.

*What is "purpose-prepared sample material"?* – it is a natural sample (e.g. dune sand) prepared for the purpose of becoming beta-source calibration material. Here it is annealing and repeated irradiation and read-out using blue-light stimulation in order to sensitise the quartz.
* * *
**Reviewer 2**

We acknowledge the need for saying more about source uniformity in the text. A description of the sources in terms of (unknown) homogeneity and shape of the radiation field is now added (lines 69ff).

*It would be very helpful to the reader if this information was given for each instrument and source used in the present study, since it would allow more meaningful interrogation of the data presented. For example, does the "open ring" have a larger active face relative to the size of the sample holder than for other sources? If so, this might explain why aliquot size does not appear to be an important consideration for this source.* Done (lines 69ff).

*The number of aliquots used to produce each datapoint in the experimental datasets is often rather low. For example, only two 8 mm diameter aliquots of R108_4 were measured using the lexsyg*

*SMART closed ring source….. I accept that the calibration samples used are bright and highly reproducible, but reliance on very small numbers of aliquots makes the resulting dataset prone to other sources of uncertainty e.g. non-uniform aliquot preparation, grain(s) between the sample and hotplate, deformed sample holders (Duller et al., 2000) etc etc. The authors should at least comment on why they think that the small number of aliquots measured doesn't pose a problem.* Uncertainties listed were actually not statistical errors suitable for comparison. We have re-calculated all errors and listed these in Table 3. Obviously, errors are big and would only become small when increasing $n$. On the other hand, the accuracy of the value will not change with $n>20$ due to the excellent reproducibility of the calibration quartz. Thus, error bars are not plotted in Figs 3 and 4.

*Three aliquot diameters were measured for sample R113_180 using the lexsyg SMART closed ring source, yet only data for the 5 mm diameter aliquots are provided in Table 2.* Apologies – correct numbers now added.

*In Figure 3a, the caption states that all data are for the 8 mm diameter aliquots, yet this size wasn't measured for sample F14_90 using the lexsyg SMART closed ring source, and the data presented appear to be for the 5 mm aliquots.* Thanks – now clarified in the caption

*Similarly, the data in Figure 3b are normalised to the 8 mm diameter aliquots, but I suspect that the 5 mm datapoint for F14_90 closed ring source is the 5 mm data normalised to itself, which is misleading.* The caption now states that data were normalised to 8mm and 5mm aliquot data.

*Figure 4 is the same as Figure 3A* – apologies. The correct figure is now inserted.

*The data in Figure 7 claim to be normalised to the 10m grain size and 7.95 mm (reported as 8 mm elsewhere in the paper) aliquot diameter, whereas I suspect they are normalised to the 10 m, 5 mm aliquot* We use 7.95 mm aliquot size for experimental and simulation data because this is the inner diameter of the cup (minus rim) which is entirely covered for fine grained samples. '8mm' is now removed from the text

*why (and how?) did you perform the GEANT4 simulation without the sample cup?* We have changed the text (lines 230f)

*don't really understand the paragraph starting on line 234, which is problematic because it appears to be critical to your explanation of grain size effects with small aliquots. Please expand and clarify.* We have changed the text (line 250ff).

*Line 65: The phrase "ring shaped source closed to the top" is awkward.* We have amended the text in sec 2.1 where sources are described. The diagrams are available through the references.

*Figure 1. Please make the small text larger.* Done

*Line 99: Please explain the rationale for using different stimulation powers for different sized aliquots and show which power was used for which size.* Now explained in the table header

*If the reduced density [1.8 g cm⁻³] is a modeller's approximation of sand deposited as a monolayer on a sample holder, please explain the logic* yes, this is correct. We have changed the text accordingly (lines 130-122)

*Figure 2: Please add A and B to the figures.* Done

*Please also provide a clearer explanation of exactly what Figure 2B is simulating* – Fig. 2B shows seven grains in plan view; they represent spheres of $SiO_2$ as stated in the caption

*Line 153: …but it would be better to be specific e.g. 4-11 μm,* a text is now added which describes the terms 'coarse' and 'fine' grain aliquot (lines 79ff)

*move the "(fg=fine grain)" statement somewhere else since it is currently out of place.* Thanks, done

*Table 3 and 4: Dashed and solid horizontal lines appear more or less at random.* Please standardise Done.

*you state that for 8 mm aliquots the effect of grain size is insignificant. By what criterion? If by your own stated in Line 171, please say so.* Not sure what is meant here. The previous paragraph does outline the criteria for significance

*Line 181: "6-26%" should read "0.6-26%".* Thanks, it should actually be 0.4 % (line 192)

*Line 182: ". . .the magnitude of the difference is controlled by the shape of the source". This sentence requires more explanation.* We feel the sentence is clear: the magnitude is the size of the difference and this size is small for the open-ring source (e.g., 0.4%) and big for the closed ring source (26%)

*Sentence starting on Line 182*…: now clarified with the correct Fig 4 displayed.

*I'm not convinced that the Armitage and Bailey (2006) data "jump" between 50 and 100 μm* .. A&B (2005) themselves identify a significant difference and say: "…*with the dose rate to 4–11 μm grains being ∼ 12% lower than that for 55–250 μm grains.*" We call this 12% difference a "jump".

*As the authors suggest (line 191), the experimental dataset is actually rather similar to the simulated data.* This is not quite what we say. Our sentence is : "There is a striking similarity between the simulated data and the experimental data adopted from Armitage and Bailey (2005), but the simulation shows a gradual change of the grain-size effect, while the experiment indicates a "jump"…"

*Line 224: I think this should read ">5%" not "<5%".* No, <5micron is correct. The drop of dose rate when grains are big (>200 micron) and aliquots are small (<5mm).

*Line 239: The figure reference should probably be to Fig. 9, in which case a reduction in the absorbed dose of ~3% looks more appropriate by eye.* The Fig reference was actually correct, but the sentence was incomplete. The text has changed (lines 250ff).

---

## Author Response (AR2)

**Notes for Revision 2**

All points raised by the reviewers have been addressed and marked up in red in the manuscript file. The following points required a statement (here below in blue):

**Reviewer#2**

Line 193: For large coarse grains Table 3 suggests that there is a (statistically insignificant) increase in dose rate with decreasing aliquot size for both the planar and closed ring source rather than a decrease as you suggest.
In the light of the additional information provided for the open-ring source (lines 70ff) the text here had to change. We now say that decreasing aliquot size increases the dose rate for all source types (line 197f).

Figure 4: The data point for F14_90 should be at ~1.04 rather than 1.08 following the correction to Table 4 noted above (I think you used the incorrectly calculated normalised % to plot Fig. 4).
I am not sure about this comment. For F14_90 the 1mm aliquots size value normalised to the 5mm value is 1.0778 derived from 0.1800/0.1670.

Line 254: By "the thicker the aliquot" I presume you mean "the larger the grain size". If so, I think the latter formulation is clearer.
No, we do indeed mean thickness. The text says: "...the smaller the aliquot, the larger the percentage of escaping secondary electrons. Furthermore, the thicker the aliquot, the smaller the percentage of secondary electrons escaping by the aliquot surface while the escape pathway via the edge remains the same."

**Reviewer#1**

l. 70: Please add here that the pre-selection of mini-sources occurs only for sources in the lexsyg research readers. The information that smart readers do not contain pre-selected sources appears only later in the text.
The text did actually say exactly this. I have re-phrased the sentence (line 69).
l. 75f.: It should read "source–sample distance". Additionally, for which grain size is this distance valid?
Thanks for pointing this out. 6.9 mm is the distance between source and surface of the sample holder – now clarified in line 74. The dose absorbed by a 240 µm grain is about 3.3% bigger than that absorbed by a 7 µm grain. This information is now added in the discussion about grain size (line 195f).

l. 181: The text describes a maximum difference of 26% between dose rates derived from various aliquot and grain sizes, but this value is not reflected by the ratio values shown in Table 4 (maximum variation of 21% in row 6 for 1 mm aliquots).
26% difference appears in Table 3 where the dose rate is normalised to the aliquot size, not to the grain size. This information is now added in the Table 3 header.

l. 194f.: The same seems to be true not only for the planar source, but also for the closed ring source (see Fig. 4).
Thanks for pointing this out. In the light of the additional information provided for the open-ring source (lines 70ff) the text here had to change. We now say that decreasing aliquot size increases the dose rate for all source types (lines 193ff).

l. 222: Please indicate that this sentence refers to experimental data. Considering the previous sentence, this is not very clear.
Thanks for pointing this out. Also here the text had to change in the light of the additional information provided for the open-ring source (lines 70ff). We now say that also the homogeneity of the source is reflected in the experimental data.